# In-domain representation learning
# for remote sensing

## Abstract

Given the importance of remote sensing, surprisingly little attention has been paid to it by the representation learning community. To address it and to establish baselines and a common evaluation protocol in this domain, we provide simplified access to 5 diverse remote sensing datasets in a standardized form. Specifically, we investigate in-domain representation learning to develop generic remote sensing representations and explore which characteristics are important for a dataset to be a good source for remote sensing representation learning. The established baselines achieve state-of-the-art performance on these datasets using in-domain representations.

## 1 Introduction

Remote sensing via computer vision and transfer learning is an important domain to address climate change as outlined by Rolnick et al. (2019). Among others, research in remote sensing promises to help in solving challenges in food security (precision farming), water sustainability, disaster prevention (floods/landslides/earthquake forecasting), deforestation or wild fire detection, urban planning, and monitoring of carbon stocks and fluxes or the air quality.

The number of Earth observing satellites is constantly increasing, with currently over 700 satellites monitoring many aspects of the Earth's surface and atmosphere from space, generating terabytes of imagery data every day. However the ground truth data acquisition is costly, usually requiring extensive campaign preparation, people and equipment transportation, and in-field gathering of the characteristics under question.

While there are remote sensing communities working on applying general deep learning methods to remote sensing problems, this domain has received relatively little attention from the representation learning community. Given its importance, it is still in an early development stage in comparison to the progress made in representation learning on natural and medical images (eg. by Raghu et al. (2019)).

Some reasons for this are the diversity of data sources (satellite types, data acquisition modes, resolutions), the need of domain knowledge and special data processing, and the wide and scattered field of applications. The scarcity of standard recognized benchmark datasets and evaluation frameworks is another.

For a long time there were only small labeled remote sensing datasets available. Only recencnly new large-scale datasets (eg. by Zhu et al. (2018); Sumbul et al. (2019)) have been generated for remote sensing problems. However, a consistent evaluation framework is still missing and the performance is usually reported on non-standard splits and with varying metrics, making reproduction and quick research iteration difficult.

To address this, we provide five representative and diverse remote sensing datasets in a standardized form for easy reuse. In particular, we explore the importance of in-domain representation learning for remote sensing at various data sizes and establish new state-of-the-art baseline results. The main goal of this work is to develop general remote sensing representations that can be applied by researchers to other unseen remote sensing tasks.

By providing these standardized datasets, common problem definition and baselines, we hope this work will simplify and enable faster iteration of research on remote sensing and inspire general representation learning experts to test their newest methods in this critical domain.

In summary, the main contributions of this work are as follows:

1. Exploring in-domain representation learning to train generic remote sensing representations.

2. Generating 5 existing remote sensing datasets in a standardized format and establishing a common evaluation protocol. Publishing the best trained representations for easy reuse in transfer learning applications[1].

3. Establishing state-of-the-art baselines for the BigEarthNet, EuroSAT, RESISC–45, So2Sat, and UC Merced datasets.

## 2 RELATED WORK

### REPRESENTATION AND TRANSFER LEARNING

Shortly after AlexNet (Krizhevsky et al., 2012) was trained on ImageNet (Deng et al., 2009), representations obtained from it were used as off-the-shelf feature extractor networks (Razavian et al., 2014). Fine-tuning approaches were also explored and led to better results than random-initialization even when the tasks or domains differ (Yosinski et al., 2014; Kornblith et al., 2019). These approaches extend beyond natural images and have been used the in medical domain, leading Raghu et al. (2019) to question on how they work. Simultaneously there have been many improvements on training representations. Mahajan et al. (2018) explored training bigger models on larger datasets with weaker labels and (Ngiam et al., 2018; Cui et al., 2018) showed gains by closer matching the domains. Very promising are the more sample-efficient approaches using semi- or self-supervision (Zhai et al., 2019a).

### DEEP LEARNING IN REMOTE SENSING

Common remote sensing problems include land-use and land cover (LULC) classification, physical and bio-geo-chemical parameter estimation, target detection, time-series analysis, pan-sharpening and change detection. Many of these tasks can be solved or helped by deep learning approaches related to classification, object detection, semantic segmentation, or super-resolution. Reviews of some of these approaches can be found for instance in (Ball et al., 2017; Zhu et al., 2017; 2019).

Remote sensing data is acquired in different modes: optical, multi- and hyper-spectral, synthetic aperture radar (SAR), lidar, spectrometer (Elachi & Van Zyl, 2006). Each of these modes has its own acquisition geometry and specific characteristics providing unique and complimentary information about the Earth's surface. In dependence of the instrument, the remote sensing imagery is available from very high resolutions at centimeter scale (aerial optical or radar, eg. for urban monitoring) to very low resolution at kilometer scale (eg. for atmosphere and ocean surface monitoring). Another important satellite data characteristic is the revisit time (how fast does a satellite revisit the same location, and constructs a time series), which can range from daily to multiple months.

The majority of deep learning approaches in remote sensing are currently based on optical imagery at high (0.3-1 m) to medium (10-30 m) resolution, obtained from aerial imagery (such as seen on Google Earth) or publicly available satellite data (eg. NASA's Landsat or ESA's Sentinel-2 satellites), respectively. Though, multi-spectral, hyper-spectral, and radar imagery is increasingly being used as well.

### REPRESENTATION/TRANSFER LEARNING IN REMOTE SENSING

Since labeling remote sensing data is expensive, for a long time there was no equivalent to ImageNet and most benchmark datasets were small. The probably most used remote sensing benchmarking dataset, UC Merced (Yang & Newsam, 2010), has only 2100 images with 100 images per class.

---

[1]To be released upon publication.

Because it was not possible to train large state-of-the-art network on a small dataset from scratch, remote sensing researchers started to build smaller networks, as for instance in Luus et al. (2015).

Another direction was to use models pre-trained on natural images. For instance, Marmanis et al. (2016) used the extracted CNN features from the pre-trained Overfeat model (Sermanet et al., 2013) to feed to another CNN model. Castelluccio et al. (2015) used GoogLeNet (Szegedy et al., 2015) pre-trained on ImageNet to fine-tune UC Merced. Similarly, Nogueira et al. (2017) used pre-trained models on natural imagery to fine-tune five models based of cross–validation splits. Afterwards, they trained an additional SVM on the fine-tuned features to achieve best performance.

Although there are works in using pre-trained representations, there is hardly any work on training representations specific for solving tasks on this domain. An example is the multi-layer generative adversarial network (GAN) (Goodfellow et al., 2014) approach for unsupervised representation learning presented by Lin et al. (2017). Xie et al. (2015) developed a transfer learning pipeline for mapping poverty distribution based on transfering the nighttime lights prediction task. Not specifically addressing representation learning, but a still related application using conditional GANs (Mirza & Osindero, 2014) was demonstrated for cloud removal from RGB images by fusing RGB and near infrared bands in Enomoto et al. (2017), or by fusing multi-spectral and synthetic aperture radar data (Grohnfeldt et al., 2018).

## 3    DATASETS

For this work, five diverse datasets were selected. We prioritized newer and larger datasets that are quite diverse from each other, address scene classification tasks, and include at least optical imagery.

Image data comes either from aerial high-resolution imagery or from satellites. Three datasets include imagery from the European Space Agency's (ESA) Sentinel-2 satellite constellation, that provides medium resolution imagery of the Earth's surface every three days. The multi-spectral imager on Sentinel-2 delivers next to the 3 RGB channels additional channels at various frequencies (see Appendix A.1). One dataset includes co-registered imagery from a dual-polarimetric synthetic aperture radar (SAR) instrument of ESA's Sentinel-1.

Besides the differences in data sources, number of training samples, number of classes, image sizes and pixel resolutions (summarized in Table 1), the datasets are also quite diverse across:

- Intra- and inter-class visual diversity: some datasets have high in-class and low between-classes diversity and vice versa.
- Label imbalance: some datatasets are perfectly balanced, while others are highly unbalanced.
- Label domain: land-use land cover (LULC), urban structures, fine ecological labels.
- Label quality: from fine human selection to weak labels from auxiliary datasets[2].

While having only 5 datasets might not allow us to completely disentangle the confounding factors of remote sensing representation learning, it should still help us in understanding the important factors for good remote sensing representation learning datasets.

### 3.1    INDIVIDUAL DATASETS

In addition to the descriptions of the individual datasets in this section, Appendix A shows example images for all datasets and the distribution of classes.

**BigEarthNet** (Sumbul et al., 2019) is a challenging large-scale multi-spectral dataset consisting of 590,326 image patches from the Sentinel-2 satellite. 12 frequency channels (including RGB) are provided, each covering an area of $1.2 \times 1.2$ km with resolutions of 10 m, 20 m, and 60 m per pixel. This is a multi-label dataset where each image is annotated by multiple land-cover classes. The label distribution is highly unbalanced ranging from 217k images of "mixed forest" label to only 328 images with label "burnt areas". About 12% of the patches are fully covered by seasonal

---

[2]Some datasets and images are affected by not filtered-out cloud and snow coverage that makes the correct classification of the samples difficult.

Table 1: Overview of considered remote sensing datasets.

| Name | year | Source | Size | Classes | Image size | Resolution | Problem |
|------|------|--------|------|---------|------------|------------|---------|
| BigEarthNet | 2019 | Sentinel-2 | 590k | 43 | 120x120* | 10–60 m | multi-label |
| EuroSAT | 2019 | Sentinel-2 | 27k | 10 | 64x64 | 10 m | multi-class |
| RESISC–45 | 2017 | aerial | 31.5k | 45 | 256x256 | 0.2–60+ m | multi-class |
| So2Sat | 2019 | Sentinel-1/2 | 376k | 17 | 32x32 | 10 m | multi-class |
| UC Merced | 2010 | aerial | 2.1k | 21 | 256x256 | 0.3 m | multi-class |

*Image size varies in dependence of resolution from 120x120 to 60x60 to 20x20.

snow, clouds or cloud shadows. The only available public baseline metrics include precision and recall values of 69.93% and 77.1%, respectively, for using a shallow CNN on a reduced dataset, after removing the snow and cloud affected samples.

**EuroSAT** (Helber et al., 2019) is another recently published dataset containing 27,000 images from Sentinel-2 satellites. All 13 frequency bands of the satellite are included. Each image covers an area of $640 \times 640$ meters and is assigned to one of 10 LULC classes, with 2000 to 3000 images per class. Because the classes are quite distinctive, very high accuracies can be achieved when using the entire dataset for training.

**NWPU RESISC–45** (Cheng et al., 2017) dataset is an aerial dataset consisting of 31,500 RGB images divided into 45 scene classes. Each class includes 700 images with a size of $256 \times 256$ pixels. This is the only dataset with varying spatial resolution ranging from 20 cm to more than 30 meters. The data covers a wide range of countries and biomes. During the construction, the authors paid special attention to have classes with high same-class diversity and between-class similarity to make it more challenging.

**So2Sat LCZ-42** (Zhu et al., 2018) is a dataset consisting of co-registered SAR and multi-spectral $320 \times 320$ m image patches acquired by the Sentinel-1 and Sentinel-2 remote sensing satellites, and the corresponding local climate zones (LCZ) (Stewart & Oke, 2012) labels. The dataset is distributed over 42 cities across different continents and cultural regions of the world. This is another challenging dataset and it is intended for learning features to distinguish various urban zones. The challenge of this dataset is the relatively small image size ($32 \times 32$) and the relatively high inter-class visual similarity.

**UC Merced** Land-Use Dataset (Yang & Newsam, 2010) is a high-resolution (30 cm per pixel) dataset that was extracted from aerial imagery from the United States Geological Survey (USGS) National Map over multiple regions in the United States. The $256 \times 256$ RGB images cover 21 land-use classes, with 100 images per class. This is a relatively small datasets that has been widely benchmarked for remote sensing scene classification task since 2010 and for which nearly perfect accuracy can be achieved with modern convolutional neural networks (Castelluccio et al., 2015; Marmanis et al., 2016; Nogueira et al., 2017).

## 3.2 RELEASE OF STANDARDIZED DATASETS

To simplify access to the data and its usage, we imported and published these datasets[3].

For reproducability and a common evaluation framework, standard *train, validation*, and *test* splits using the 60%, 20%, and 20% ratios, respectively, were generated for all datasets except So2Sat.

For the So2Sat dataset, the source already provides train and validation splits. To generate the test split, the original upstream validation is separated into validation and test splits with the 25% and 75% ratios, respectively.

---

[3]To be updated/released upon publication.

## 4 Remote Sensing Data Processing

The remote sensing domain is quite distinctive from natural image domain and requires special attention during pre-processing and model construction. Some characteristics are:

- Remote sensing input data usually comes at higher precision (16 or 32 bits).
- The number of channels is variable, depending on the satellite instrument. RGB channels are only a subset of a multi- or hyper-spectral imagery dataset. Other data sources might have no optical channels (eg. radar or lidar) and the channels distribution can be determined by polarimetric, interferometric or frequency diversity.
- The range of values varies largely from dataset to dataset and between channels. The values distribution can be highly skewed.
- Many quantitative remote sensing tasks rely on the absolute values of the pixels.
- The images acquired from space are usually rotation invariant.
- Source data can be delivered at different product levels (for instance w/ or w/o atmospheric correction, co-registration, orthorectification, radiometric calibration, etc.).
- Especially lower resolution data aggregates a lot of information about the illuminated surface in a single pixel since it covers a large area.
- Image axes might be non-standard, eg. representing range and azimuth dimensions.

This sets some requirements on pre-processing and encourages to adjust data augmentation of the input pipeline for remote sensing data.

Specifically for the problems discussed in this paper, it is recommended to rescale and clip the range of values per channel (accounting for outliers). Data augmentation that affects the intensity of the values should be discarded. On the other hand, one can reuse the rotation invariance and extend the augmentation to perform all rotations and flipping (providing 7 additional images per sample). Given multi-spectral data, such as Sentinel-2 based BigEarthNet, EuroSAT and So2Sat, one can use other subsets of channels instead of RGB including all available ones.

## 5 Experimental Results

The main goal is to develop representations that can be used across a wide field of remote sensing applications on unseen tasks. The training and evaluation protocol follows two main stages: (1) *upstream* training of the representations model based on some out- or in-domain data, and (2) *downstream* evaluation of the representations by transferring the trained representation features to the new downstream tasks. For the upstream training on in-domain proxy datasets the entire data is used, that cannot include any data from the downstream tasks. The quality of the trained representations and their generalizability is often evaluated on reduced downstream training sizes to assess efficiency of the representations.

### 5.1 Experimental setup

In order to simplify the training procedure and to draw more general conclusions, all experiments use the same ResNet50 V2 architecture (He et al., 2016) and configuration. However, due to the varied number of classes and training samples in the various datasets, we perform sweeps over a small set of hyper-parameters and augmentations as described in detail in Appendix B.

### 5.2 Evaluation metrics

We report performance results using accuracy metrics commonly used in computer vision tasks: for multi-class problems we use the Top-1 global accuracy metric, which denotes the percentage of correctly labeled samples. For multi-label problems we use the mean average precision (mAP) metric, which denotes the mean over the average precision (AV) values (AV is the integral over the precision-recall curve) of the individual labels.

Table 2: Performance of trained In-Domain and ImageNet representations (rows) when using 1000 training examples for downstream tasks (columns). Emphasized (bold font) are the best accuracies per downstream task (column).

| Source \Target | BigEarthNet | EuroSAT | RESISC–45 | So2Sat | UC Merced |
|---|---|---|---|---|---|
| ImageNet | 25.10 | 96.84 | 84.89 | 53.69 | 99.02 |
| BigEarthNet | - | 96.45 | 78.43 | 50.91 | **99.61** |
| EuroSAT | 27.10 | - | 79.59 | 52.99 | 98.05 |
| RESISC–45 | **27.59** | **97.14** | - | **54.43** | **99.61** |
| So2Sat | 26.30 | 96.30 | 77.70 | - | 97.27 |
| UC Merced | 26.86 | 96.73 | **85.73** | 53.52 | - |

To measure and aggregate relative performance improvement over datasets performing at quite different accuracy levels, the logit transformation of the accuracy is preferred (Kornblith et al., 2019):

$$\Delta = \text{logit}(\rho) = \log\left(\frac{\rho}{1-\rho}\right) = \frac{1}{\text{sigmoid}(\rho)}$$

where $\rho$ is the accuracy rate (Top-1 or mAP) and $\Delta$ is the corresponding logit accuracy. It captures the importance of accuracy change at different accuracy levels to provide a more fair evaluation of the achievement.

### 5.3 Comparing in-domain representations

To obtain in-domain representations, first we train models either *from scratch* or by *fine-tuning* ImageNet on each full dataset. The best of these models are then used as in-domain representations to train models on other remote sensing tasks (excluding the one used to train the in-domain representation).

For an initial evaluation of the different in-domain representation source data, Table Table 2 shows a cross-table of evaluating each trained in-domain and ImageNet representation on each of the downstream tasks. The representations were trained using full datasets upstream, while the down-stream tasks used only 1000 training examples to better emphasize the differences.

It can be observed that with 1000 training examples, the best results all come from fine-tuning the in-domain representations.

Additionally, despite having 2 distinctive groups of high-resolution aerial (RESISC–45, UC Merced) and medium-resolution satellite datasets (BigEarthNet, EuroSAT and So2Sat), the representations trained on RESISC–45 were able to outperform the others in all tasks (BigEarthNet representations tied for the UC Merced dataset) and it was the only representation to consistently outperform ImageNet-based representations.

That RESISC–45 would perform so good on both aerial and satellite data was unexpected. The reason is most likely related to the fact that RESISC–45 is the only dataset that has images with various resolutions. Combined with the large number of classes that have high within class diversity and high between-class similarity it seems to be able to train good representations for a wide range of remote sensing tasks, despite not being a very big dataset. This learning can be reused to adjust augmentation schemes for remote sensing representation learning by stronger varying the scale of images with randomized zooms and potentially other affine transformations.

The best representations for the RESISC–45 itself come from the other aerial dataset, UC Merced. The relatively small training size of it was counter-balanced by an aggressive augmentation (*crop & rot* as described in Appendix B.3).

Counter to the expectation that bigger datasets should train better representations, the two biggest datasets, BigEarthNet and So2Sat, didn't provide the best representations (except of BigEarthNet representations for UC Merced). We hypothesize that this might be due to the weak labeling and the low training accuracy obtained in these datasets. It is possible that the full potential of these large-scale datasets was not yet fully utilized and other self- or semi-supervised representation learning approaches could improve the performance.

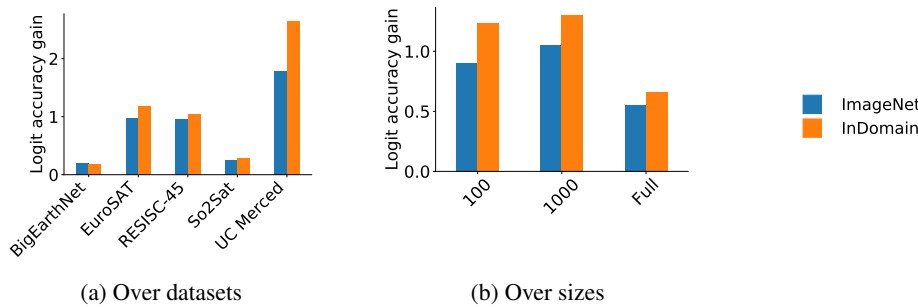

(a) Over datasets       (b) Over sizes

Figure 1: Aggregated mean relative improvement in logit accuracy of fine-tuning from ImageNet and in-domain representations in comparison to training from scratch.

## 5.4 LARGE-SCALE COMPARISON

Having trained in-domain representations, we can now evaluate and compare the transfer quality of fine-tuning the best in-domain representations with fine-tuning ImageNet and training from scratch at various training data sizes. The results using the default experimental setup are shown in Table 3.

In all cases, fine-tuning from ImageNet is better than training from scratch. And in all but one case, fine-tuning from an in-domain representation for transfer is even better.

The only exception is the BigEarthNet dataset at its full size. It is expected that having a large dataset should reduce the need for pre-training, but the gap between in-domain and ImageNet pre-training is quite big. We don't have an explanation for this yet and this needs to be further investigated.

Overall, these results establish new state-of-the-art baselines for these datasets, as summarized in Table 4. Note that some results are not comparable: RESISC–45 has been previously evaluated only on 20% of data, So2Sat has no public benchmarking result to our knowledge, and the only published result of BigEarthNet is based on a cleaner version of the dataset (after removing the noisy images containing clouds and snow) and only precision and recall metrics were reported.

Fig. 1 summarizes the results of Table 3 across all datasets and training sizes. Logit accuracy metric is used, which fits better for aggregation of accuracies across wide ranges. Fig. 1a emphasizes that the smaller datasets (EuroSAT, RESISC–45, UC Merced) profit more from *in-domain* knowledge than the larger datasets (BigEarthNet, So2Sat). As is re-iterated again in Fig. 1b, using in-domain representations leads to higher accuracy gain when small number of training examples are available, which is of most interest for remote sensing applications to reduce the ground truth data acquisition.

## 5.5 LIMITED NUMBERS OF TRAINING EXAMPLES

To look closer into in-domain representation learning for small number of training examples, we trained models with small training sizes ranging from 25 to 2500 (samples were randomly drawn disregarding class distributions). We used a simplified set of hyper-parameters that might not deliver the most optimal performance, but still allows to observe the general trends. As shown in Fig. 2, the improvement of using in-domain representations is clearly visible for the EuroSAT, RESISC–45 and UC Merced datasets. These are the 3 smaller datasets with higher quality labels. The results are less conclusive for the BigEarthNet and So2Sat datasets that have more noisy labels.

Table 3: Accuracy over different training methods and number of used training samples.

|  | BigEarthNet | | | EuroSAT | | | RESISC–45 | | | So2Sat | | | UC Merced | | |
|---|---|---|---|---|---|---|---|---|---|---|---|---|---|---|---|
|  | 100 | 1k | Full | 100 | 1k | Full | 100 | 1k | Full | 100 | 1k | Full | 100 | 1k | Full |
| Scratch | 14.5 | 21.4 | 72.4 | 63.9 | 91.7 | 98.5 | 21.4 | 56.1 | 95.6 | 33.9 | 47.0 | 62.1 | 50.8 | 91.2 | 95.7 |
| ImageNet | 17.8 | 25.1 | **75.4** | 87.3 | 96.8 | 99.1 | 44.9 | 84.9 | 96.6 | 44.9 | 53.7 | 63.1 | 79.9 | 99.0 | 99.2 |
| InDomain | **18.8** | **27.6** | 69.7 | **91.3** | **97.1** | **99.2** | **49.0** | **85.7** | **96.8** | **46.4** | **54.4** | **63.2** | **91.0** | **99.6** | **99.6** |

Table 4: Results on selected remote sensing datasets. All results use Top-1 accuracy, except for BigEarthNet which uses precision/recall and mean average precision. *Our* best results were obtained by fine-tuning in-domain representations except for BigEarthNet which was obtained by fine-tuning ImageNet.

| Dataset | Reference | Result |
|---|---|---|
| BigEarthNet | Sumbul et al. (2019) | 69.93% / 77.1% (P/R) |
| | *Ours* | *75.36% (mAP)* |
| EuroSAT | Helber et al. (2019) | 98.57% |
| | *Ours* | *99.20%* |
| RESISC–45 | Cheng et al. (2017) | 90.36% |
| | *Ours* | *96.83%* |
| So2Sat | *Ours* | *63.25%* |
| UC Merced | Marmanis et al. (2016) | 92.4% |
| | Castelluccio et al. (2015) | 97.1% |
| | Nogueira et al. (2017) | 99.41% |
| | *Ours* | *99.61%* |

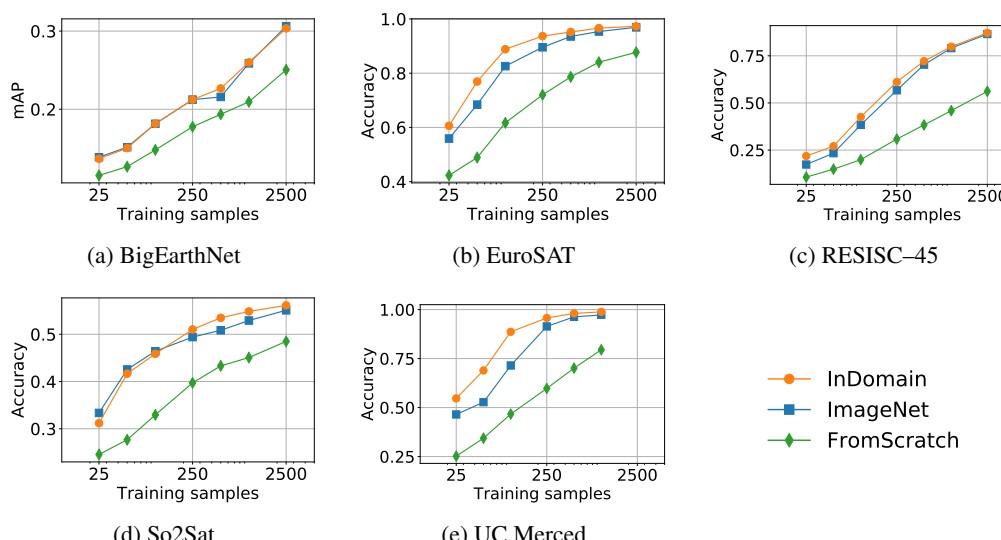

Figure 2: Top-1 accuracy rate or mean average precision (mAP) on validation set after training with a given method over a limited number of training examples on each dataset.

## 6 CONCLUSION

We present a common evaluation benchmark for remote sensing representation learning based on five diverse datasets. The results demonstrate the enhanced performance of *in-domain* representations, especially for tasks with limited number of training samples, and achieve state-of-the-art performance. The five analyzed datasets and the best trained in-domain representations are published for easy reuse by the public.

We investigate dataset characteristics to be a good source for remote sensing representation learning. As the experimental results indicate, having a multi-resolution dataset helps to train more generalizable representations. Other factors seem to be label quality, number of classes, visual similarity across the classes and visual diversity within the classes. Surprisingly, we observed that representations trained on the large weakly-supervised datasets were not as successful as that of a smaller and more diverse human-curated dataset.

However, some results were inconclusive and require more investigation. Understanding the main factors of a good remote sensing dataset for representation learning is a major challenge, solving which could improve performance across a wide range of remote sensing tasks and applications.

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

# A ADDITIONAL DATASET INFORMATION

This section provides additional information on the used datasets and data sources.

## A.1 SENTINEL-2 SATELLITE

Sentinel-2 is a multi-spectral satellite constellation from the European Space Agency (ESA). Since 2017 two satellites are in operation delivering a 5-days revisit at equator and 2-3 days at high latitudes. The characteristics of the 13 bands are presented in Table 5.

Table 5: Sentinel-2 channel characteristics (Abbreviations: NIR: Near Infra-Red, SWIR: Short-Wavelength Infra-Red).

| Band and Highest Sensitivity Target | Spatial Resolution [m] | Central Wavelength [nm] |
|---|---|---|
| B01 - Aerosols | 60 | 443 |
| B02 - Blue | 10 | 490 |
| B03 - Green | 10 | 560 |
| B04 - Red | 10 | 665 |
| B05 - Red edge 1 | 20 | 705 |
| B06 - Red edge 2 | 20 | 740 |
| B07 - Red edge 3 | 20 | 783 |
| B08 - NIR | 10 | 842 |
| B08A - Red edge 4 | 20 | 865 |
| B09 - Water vapor | 60 | 945 |
| B10 - Cirrus | 60 | 1375 |
| B11 - SWIR 1 | 20 | 1610 |
| B12 - SWIR 2 | 20 | 2190 |

## A.2 BigEarthNet

The following figures show some example images and label distribution for the BigEarthNet dataset.

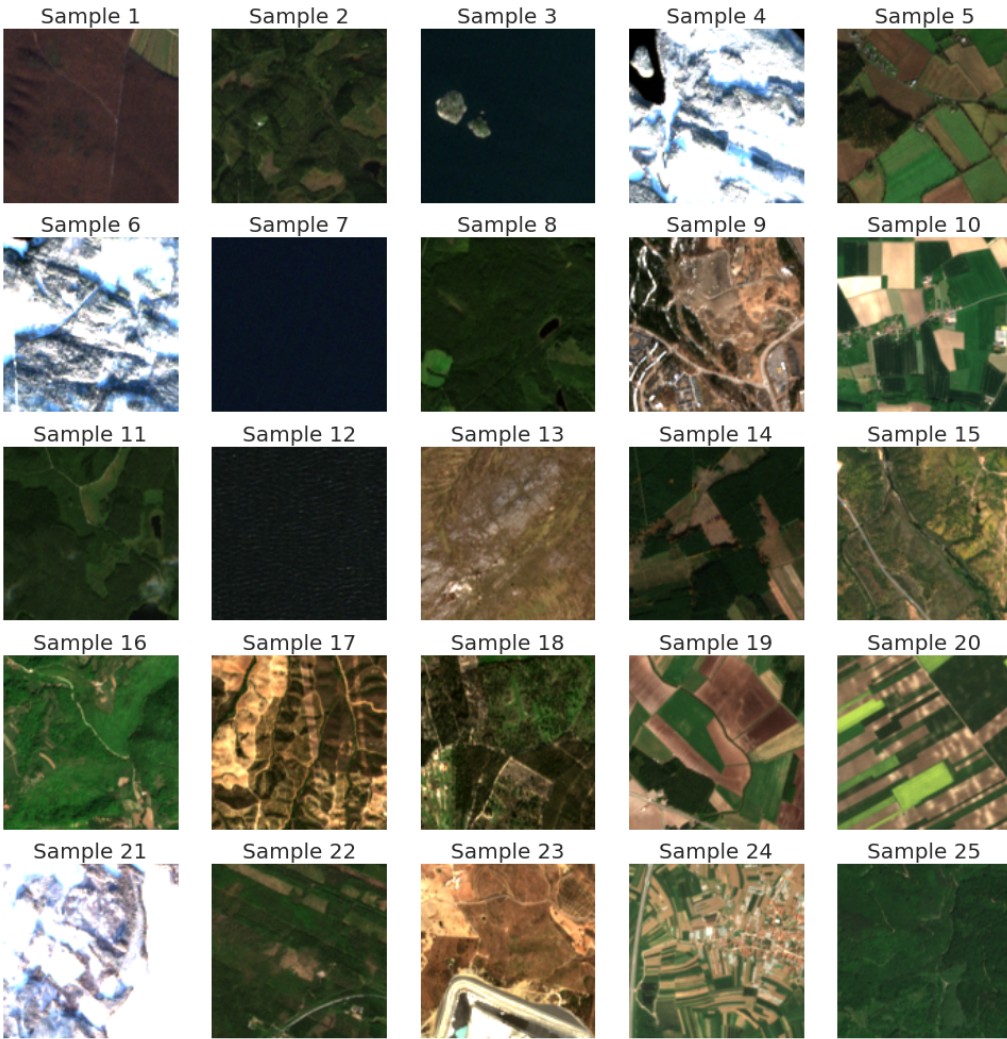

Figure 3: BigEarthNet image examples. Some images might be affected by seasonal snow, clouds or cloud shadows, which is not reflected in the land cover labels of this dataset.

Labels for the examples in Fig. 3:

1. Vineyards, Broad-leaved forest (2 labels)
2. Coniferous forest, Mixed forest, Transitional woodland/shrub (3 labels)
3. Sea and ocean (1 label)
4. Land principally occupied by agriculture, with significant areas of natural vegetation, Coniferous forest, Mixed forest, Water bodies (4 labels)
5. Non-irrigated arable land, Mixed forest (2 labels)
6. Land principally occupied by agriculture, with significant areas of natural vegetation, Coniferous forest, Mixed forest, Transitional woodland/shrub (4 labels)
7. Sea and ocean (1 labels)
8. Coniferous forest, Mixed forest (2 labels)

9. Discontinuous urban fabric, Industrial or commercial units, Coniferous forest, Mixed forest, Transitional woodland/shrub (5 labels)

10. Non-irrigated arable land, Pastures (2 labels)

11. Coniferous forest, Mixed forest, Water bodies (3 labels)

12. Sea and ocean (1 labels)

13. Sparsely vegetated areas, Peatbogs (2 labels)

14. Coniferous forest, Mixed forest, Transitional woodland/shrub (3 labels)

15. Continuous urban fabric (1 label)

16. Complex cultivation patterns, Land principally occupied by agriculture, with significant areas of natural vegetation, Broad-leaved forest (3 labels)

17. Land principally occupied by agriculture, with significant areas of natural vegetation, Broad-leaved forest, Sclerophyllous vegetation, Transitional woodland/shrub (4 labels)

18. Agro-forestry areas, Broad-leaved forest, Transitional woodland/shrub (3 labels)

19. Non-irrigated arable land, Land principally occupied by agriculture, with significant areas of natural vegetation, Coniferous forest, Mixed forest (4 labels)

20. Non-irrigated arable land (1 label)

21. Coniferous forest, Mixed forest, Transitional woodland/shrub, Water bodies (4 labels)

22. Coniferous forest, Mixed forest (2 labels)

23. Non-irrigated arable land, Land principally occupied by agriculture, with significant areas of natural vegetation, Agro-forestry areas, Sclerophyllous vegetation, Transitional woodland/shrub, Water bodies (6 labels)

24. Discontinuous urban fabric, Non-irrigated arable land, Inland marshes (3 labels)

25. Land principally occupied by agriculture, with significant areas of natural vegetation, Broad-leaved forest, Coniferous forest (3 labels)

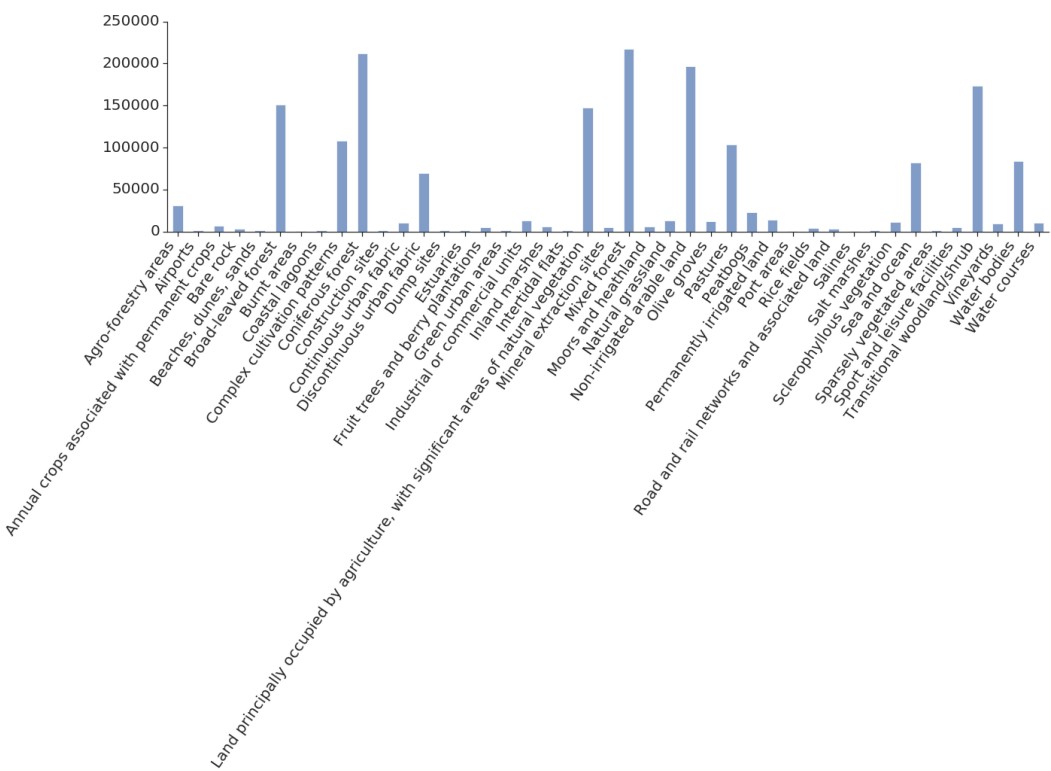

Figure 4: BigEarthNet labels distribution counts.

## A.3 EUROSAT

The following figures show some example images and label distribution for the EuroSAT dataset.

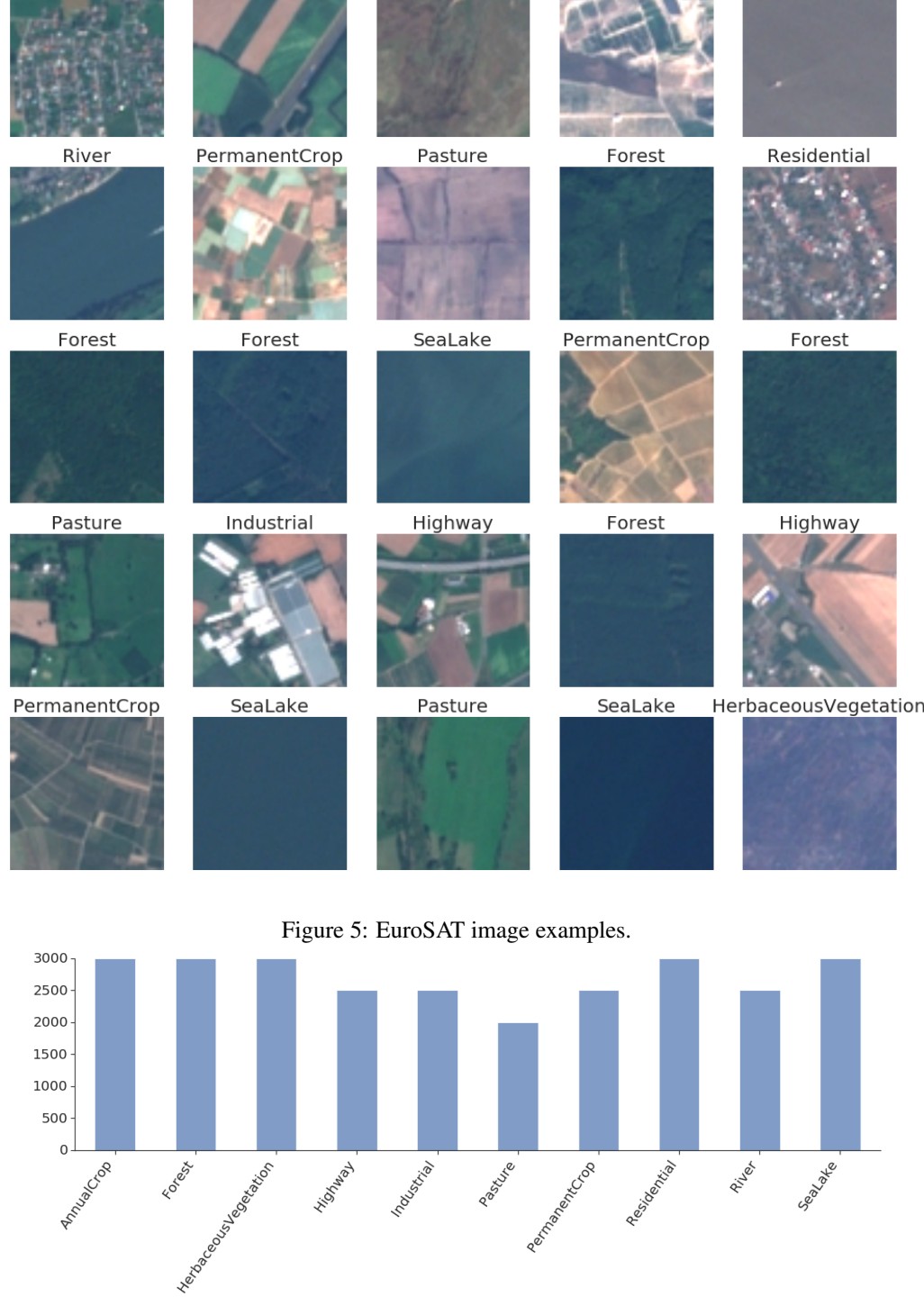

Figure 5: EuroSAT image examples.

Figure 6: EuroSAT class distribution counts.

## A.4 RESISC–45

The following figures show some example images and label distribution for the RESISC–45 dataset.

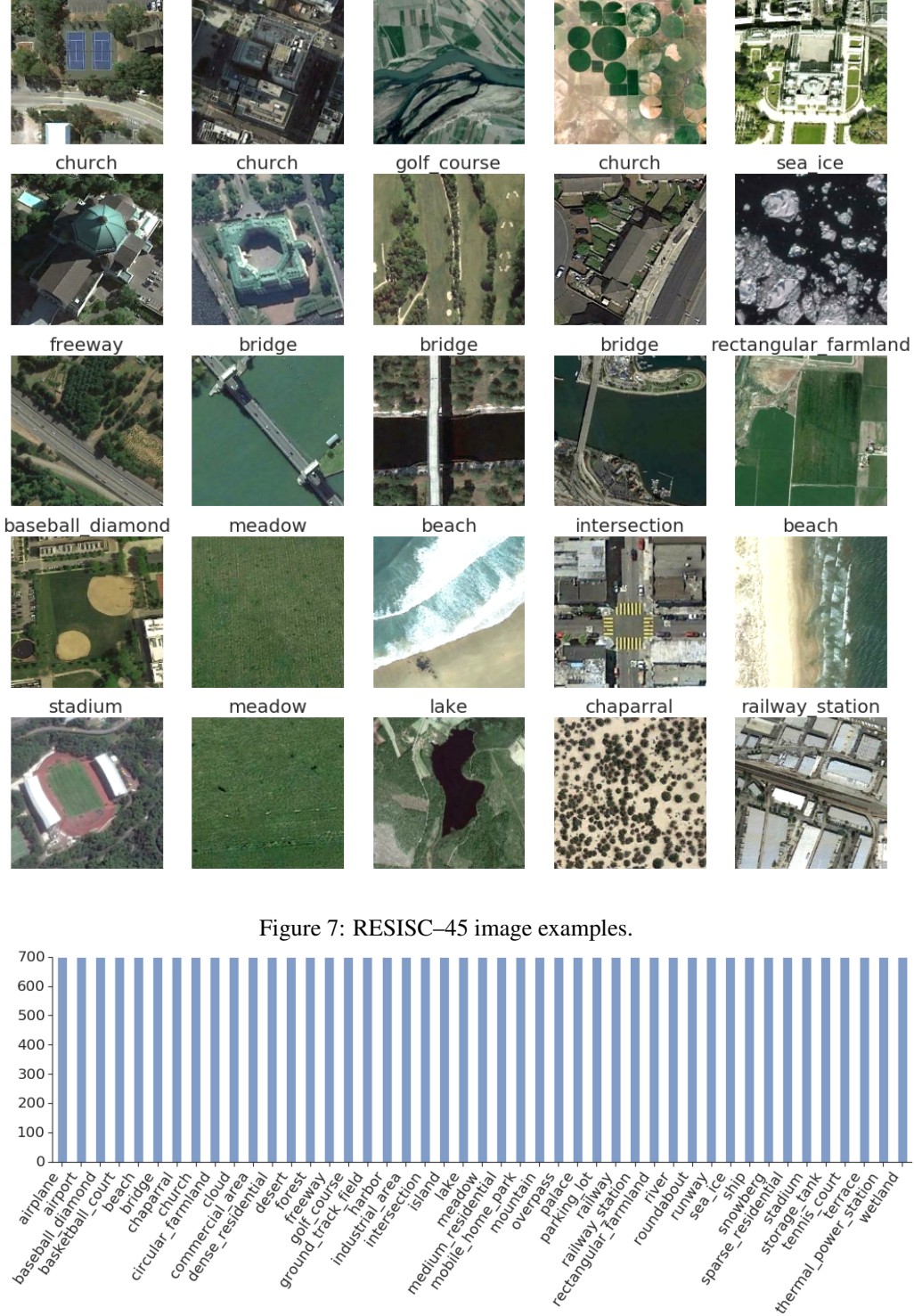

Figure 7: RESISC–45 image examples.

Figure 8: RESISC–45 class distribution counts.

## A.5 So2Sat

The following figures show some example images and label distribution for the So2Sat dataset.

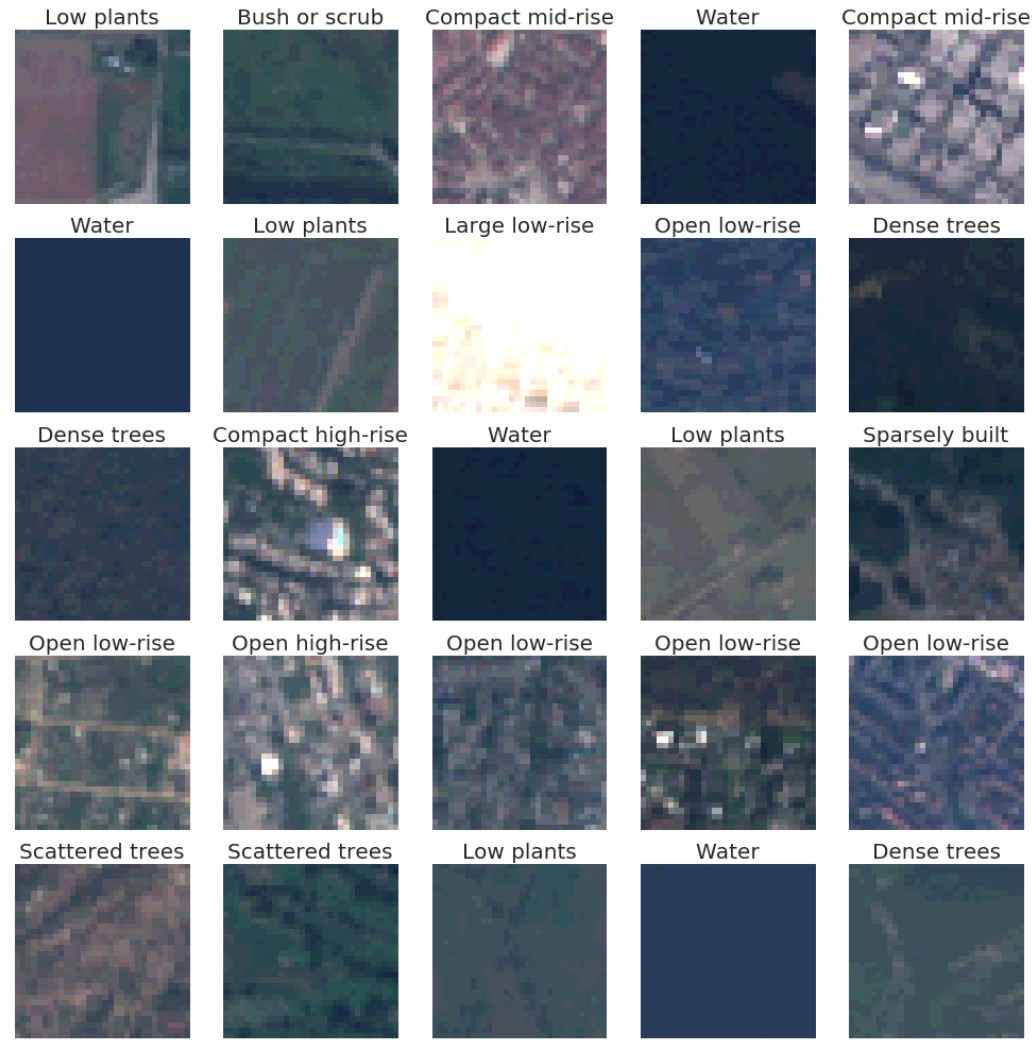

Figure 9: So2Sat image examples.

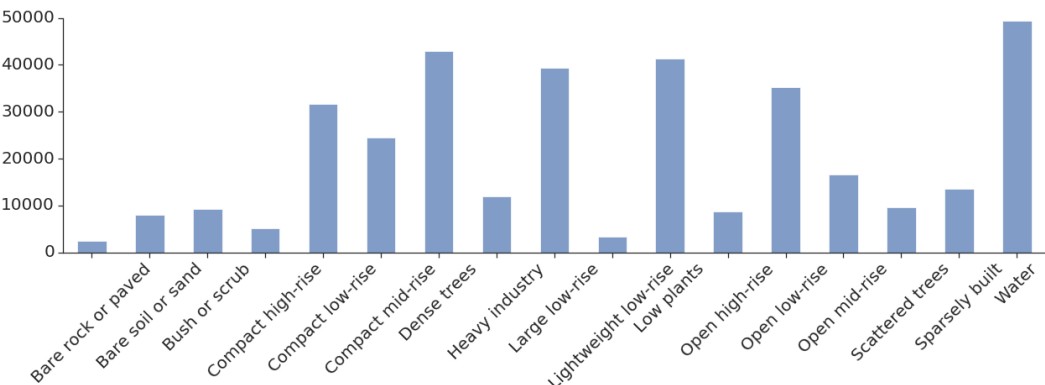

Figure 10: So2Sat class distribution counts.

### A.6 UC MERCED

The following figures show some example images and label distribution for the UC Merced dataset.

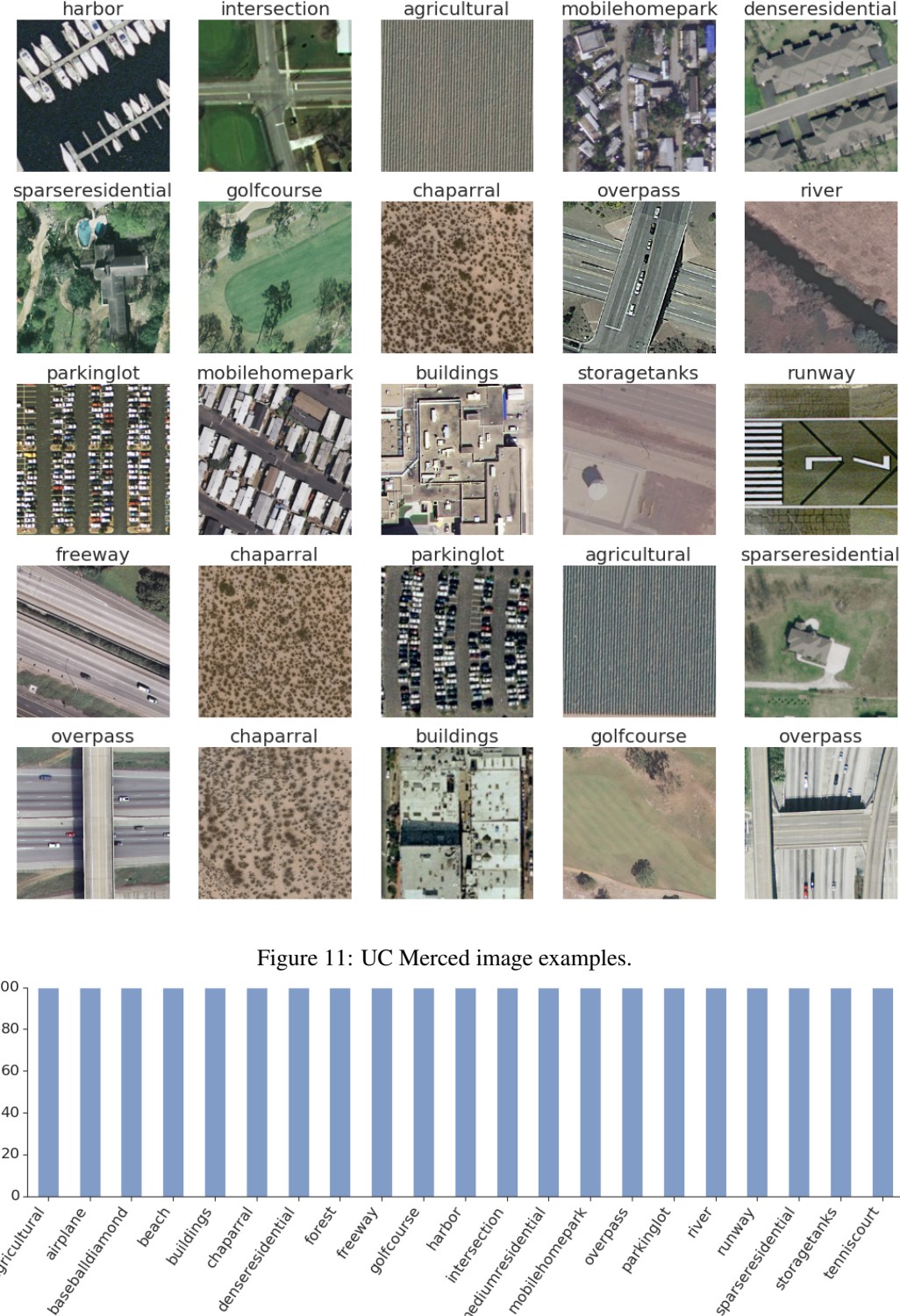

Figure 11: UC Merced image examples.

Figure 12: UC Merced class distribution counts.

## B    TRAINING SETUP

### B.1    ARCHITECTURE AND HYPER-PARAMETERS

All the models trained in this work use the ResNet50 v2 architecture (He et al., 2016) and were trained using SGD with momentum set to 0.9 on large batch sizes 512 or 1024. In total we configure and sweep only a fixed set of hyper-parameters per model, namely:

- learning rate: $\{0.1, 0.01\}$,
- weight decay: $\{0.01, 0.0001\}$,
- training schedules: $\{$short, medium, long$\}$,
- preprocessing: $\{$*resize*, *crop & rot*$\}$ (described in Appendix B.3).

All the 3 training schedules use a linear warm-up for learning rate over the first $w$ steps/epochs and decrease the learning rate by 10 per each learning phase $p$ in the schedule. The learning rate schedules are given by:

- short: $p = \{750, 1500, 2250, 2500\}$ steps, $w = 200$ steps.
- medium: $p = \{3000, 6000, 9000, 10000\}$ steps, $w = 500$ steps.
- long: $p = \{30, 60, 80, 90\}$ epochs, $w = 5$ epochs.

These hyperparameter settings follow approximately the setup in (Zhai et al., 2019b) with minor modifications for the schedule and preprocessing. In in inital phase, more extensive hyperparameter sets were tried out, but with not much effect on the best performance and therefore for the experiments presented in this paper we limit the configurations to the ones described above.

### B.2    DETAILED SWEEPS PER EXPERIMENT

The models for experiments in Table 2 and Table 3 were trained sweeping over all hyper parameters. The best performing models obtained from fine-tuning ImageNet were used as in-domain representations for all experiments (excluding same up- and down-stream dataset configurations).

For Fig. 2, the models were trained by sweeping over the learning rate and using only the short and medium training schedules. The weight decay was set to 0.0001, and the preprocessing was set to *resize*.

### B.3    PRE-PROCESSING AND DATA AUGMENTATION

Data pre-processing and augmentation can have a significant impact on performance. Therefore, in the reported results we used 2 pre-processing settings described below.

- *resize* - resize the original RGB input to 224x224 both at training and evaluation time.
- *crop & rot* - during training resize the original RGB input to 256x256, perform a random crop of 224x224 and apply one of 8 random rotations (90 degrees and horizontal flip). During evaluation resize to 256x256 and perform a central crop of 224x224.

Table 6 shows the difference of each strategy when fine-tune from ImageNet.

Table 6: Accuracy of each pre-processing strategy when fine-tuning an ImageNet representation.

|  | BigEarthNet | | | EuroSAT | | | RESISC–45 | | | So2Sat | | | UC Merced | | |
|---|---|---|---|---|---|---|---|---|---|---|---|---|---|---|---|
|  | 100 | 1k | Full | 100 | 1k | Full | 100 | 1k | Full | 100 | 1k | Full | 100 | 1k | Full |
| *resize* | 17.3 | 24.5 | **75.4** | 85.2 | 96.0 | 98.9 | 37.5 | 78.6 | 95.8 | **44.9** | 51.8 | **63.1** | 69.1 | 98.2 | **99.2** |
| *crop & rot* | **17.8** | **25.1** | 73.4 | **87.3** | **96.8** | **99.1** | **44.9** | **84.9** | **96.6** | 43.8 | **53.7** | 59.6 | **79.9** | **99.0** | **99.2** |

