# OpenReview forum: "In-Domain Representation Learning For Remote Sensing"
_ICLR.cc/2020/Conference — Reject_

### Official Review · AnonReviewer1 · 2019-10-21
**Official Blind Review #1**

**Rating:** 3

**Review:**

This paper explored the representation learning problem on remote sensing domain. In particular, it provided several standardized remote sensing data sets with standard train/validation/test splits and various metrics for fairly comprehensive evaluation. It showed that compared to fine-tuning on ImageNet or learning from scratch, in-domain representation could produce better baseline results for remote sensing at various training data sizes.
Overall, I prefer to reject this paper based on the following reasons:
(1) It claimed to address the question “what characteristics should a dataset have to be a good source
for remote sensing representation learning”. But this question has not been fully explored. Empirically, multi-resolution data sets (e.g., RESISC-45) could be used as the base data set for better in-domain representation learning. However, the essential reason on this explaining this phenomenon is not still clear.
(2) The authors did not provide the convincing evidence regarding how to generate the standard train/validation/test splits for all the data sets. The data split might be the reason to explain the superior performance of in-domain representation learning.
(3) On evaluating the in-domain representations (shown in Table 2), all the representation models were trained with 1000 training examples for performance comparison. It is not convincing whether the best in-domain representation models are affected by the training size. Later, it used the best in-domain representation models to estimate the state-of-the-art baselines on various training size. It is not clear how to select the best in-domain representation models.
Minor comments:
(1) In Section 5.5, the authors trained the model with randomly sampled training examples. It might be more convincing if they can show the mean and variance of model performance using multiple randomly sampled examples.


**Experience Assessment:**

I have read many papers in this area.

**Review Assessment: Checking Correctness Of Derivations And Theory:**

I carefully checked the derivations and theory.

**Review Assessment: Checking Correctness Of Experiments:**

I assessed the sensibility of the experiments.

**Review Assessment: Thoroughness In Paper Reading:**

I read the paper thoroughly.

---

> ### Author Response · Authors · 2019-11-14
> **Response to Reviewer #1**
>
> Thank you for the review.
> Some parts of the paper were wrongly interpreted and we have modified the content of the paper to make it clearer.
>
> To 1: With this work, we are addressing a challenging domain, providing datasets, defining common benchmarks and demonstrating state-of-the-art performance based on various approaches. To our knowledge, this is the first generic and systematic study on remote sensing representation learning based on different tasks and datasets. We addressed the mentioned question to the best of our current understanding, but we do not claim that we fully understand the entangling factors and do not consider this solved.
> Although we made the insights as discussed in the Conclusion section, we did not aim to solve representation learning for remote sensing in this one paper, but wanted to publicise this important domain to the community. And we believe the lessons and insights we learned in this domain are worth sharing and discussing. But thank you for your comment - we de-emphasized this aspect to avoid misconceptions about the intent of our paper.
>
> To 2: We want to provide standard splits so that one can use the same evaluation protocol to better compare results across the community for better reproducibility. The splits in the literature of the used datasets are often not known to us and therefore it is not possible for anybody to do a fair comparison. We discuss this when we present the datasets and the results.
> Please note that since we used the same splits for all investigated methods, there is no bias for in-domain representation learning.
>
> To 3: The representations were trained using the full datasets, but the downstream tasks were evaluated in this case using only 1000 examples. As discussed in the text, Table 2 focuses on how different in-domain representation learning datasets affect other downstream tasks, and it was shown for the case when using only 1000 training examples downstream. Note that we evaluated other downstream data sizes in Table 3 and Figure 2. For the SOTA results reported in Table 4, we used the full datasets upstream and downstream. We updated the paper to make this clearer.
>
> Concerning the best in-domain representation model, among the evaluated approaches there is one clear winner, as discussed in the paper and which we want to open-source for other researchers to try out. However, there is no guarantee that these representations would be the best for all possible downstream tasks and usually it is good practice to evaluate different ones. Representation/model selection is a separate topic of itself and is out of scope for this paper.
>
> To 4 (minor (1)): As for almost all standard ML benchmark datasets (ImageNet, Cifar, etc.) we always return the same set of samples in order to guarantee that the evaluation of different methods is performed on exactly the same data. Please note that with this work we wanted to closer align the remote sensing tasks with the common protocols used in representation learning community.

---

### Official Review · AnonReviewer3 · 2019-10-30
**Official Blind Review #3**

**Rating:** 3

**Review:**

### Summary
This paper discusses transfer learning in remote sensing image-classification tasks.
The key finding is: while transfer learning from ImageNet dataset for remote sensing, performance can be slightly improved if the model weights are fine-tuned on an auxiliary remote sensing dataset (hence in-domain transfer).
​
​
### Strengths
- The paper is well motivated for the need of good representation learning for remote sensing, and is written well making it easy to follow.
- It compiles a nice list of five datasets available for remote sensing, along with their characteristics. Also, the list of common properties of such datasets (e.g. rotation invariant) is good for data pre-processing for engineering applications in remote sensing.
- The paper provides baseline accuracies on five important datasets in remote sensing.
​
​
### Weaknesses
Apart from the contributions of compilation of different datasets which are already present, and classification results on them, there is not much novelty in the paper. There are several major weaknesses:
​
1. **About datasets**: The paper emphasizes a lot on the lack of standard datasets and metrics in remote sensing. However, the contribution is limited to compiling existing datasets and defining a 60-20-20 train-validation-test split on them. There is no new data collection, no insight in how these existing diverse datasets could be combined together, or why this particular definition of split is used. Hence, the reviewer cannot count this as a contribution.
2. **About representation learning**: Since the source domain (ImageNet), in-domain source domain and the target domain all solve classification tasks, a better term to use would be *transfer learning* instead of representation learning. The reviewer recommends to demonstrate how these learned "representations" are used for other downstream tasks or otherwise show some other representation learning methods (semi-supervised or unsupervised), which is not same as the downstream task. Hence, it would be inappopriate to sell it as representation learning, in its current form.
3. **About in-domain learning**: Since the in-domain transfer is also done by fine-tuning an existing model which was trained on ImageNet, it is expected to do somewhat better than Imagenet only. The paper just clarifies this fact in practice, and the performance gains are also statistically insignificant. Again, this is not a significant contribution.
4. **About hyperparameter sweeping**: The minor improvements observed due to in-domain transfer is also unreliable, since very hyperparameters are tried across datasets. E.g. only 2 values of learning rates (0.1 and 0.01) are tested. This could also explain why BigEarthNet and So2Sat had poor training accuracies. The reviewer also disagrees with using only 1k samples for selecting best dataset to transfer from - explained in minor weaknesses.
5. **About experiments**: In-domain transfer is tested by fine-tuning on one dataset and then again fine-tuning on another dataset. It was observed that the dataset RESISC-45 does the best in this scenario. The reviewer disagrees with paper's hypothesis about noisy labels resulting in this artifact. The reviewer suspects that this is because RESISC-45 contains labels which are most appropriate for classification transfer in other datasets (and also it is very balanced). Can the authors comment on this? If another method of representation learning than classification is employed, then it is very possible that other larger datasets result in better representations. The comparison across datasets currently is strange, since all it tells is which dataset has most similar classification labels to other datasets on average. To remove this problem in comparison, the representation learning task and testing task should be different.
6. **About conclusions and contributions**: The experimental setup in this paper is insufficient to claim that it investigates *what characteristics should a dataset have to be a good source for remote sensing representation learning*. Transfer from ImageNet works because it contains a large number of classes, along with data diversity and its sheer size. This cannot be said for the datasets analyzed in this paper, since these factors are not analyzed individually, and the representation learning method is not standardized across all datasets (classification is not standardized since different datasets have different classes), making the comparison unfair.
​
​
​
#### Minor weaknesses:
- Table 4: Since existing BigEarthNet result contains precision/recall, that is what the authors should provide for their method.
- The best dataset to transfer from is evaluated only on 1k samples, which is too small. The accuracy changes ~25% to ~70% accuracy for BigEarthNet when going from 1k samples to full dataset. Hence the trend is unpredictable, and some other dataset could have potentially performed better on full dataset.
- Remove blank page at the end
​
### Suggestions to Improve
For this to be a comprehensive survey on in-domain representation learning in remote sensing, the reviewer recommends to add:
​
1. Standardized representation learning methods to compare across datasets.
2. Results on more downstream tasks (other than classification; examples mentioned in the paper introduction) to evaluate representation learning.
​
The reviewer understands these could be out of scope of this paper. However in its current state, the contributions are not significant enough.
​


**Experience Assessment:**

I do not know much about this area.

**Review Assessment: Checking Correctness Of Derivations And Theory:**

I assessed the sensibility of the derivations and theory.

**Review Assessment: Checking Correctness Of Experiments:**

I assessed the sensibility of the experiments.

**Review Assessment: Thoroughness In Paper Reading:**

I read the paper at least twice and used my best judgement in assessing the paper.

---

> ### Author Response · Authors · 2019-11-14
> **Response to Reviewer #3**
>
> Thank you for the review and the detailed comments.
> We believe that some concepts and goals were misunderstood and we have updated the paper accordingly.
>
> To 1: Although it appears “unexciting”, creating a well defined split to standardize research and comparison is an important step in making progress and enable reproducibility of research. Our release can greatly benefit the community. The splits were selected to be consistent with similar evaluation protocols to simplify adaptation of this important domain and to enable reproducibility (something that was hardly possible before on the remote sensing datasets).
>
> To 2:  We want to develop and open-source representations that can be applied by other remote sensing practitioners on a diverse set of (unknown to us) downstream tasks and applications. As well, we want to provide a set of standardized datasets that can be used by general representation learning specialists and to explain how these differ from the usual natural image domain. The focus for us is on understanding and developing generic representations for remote sensing data domains. We as well explored other representation learning approaches including semi- and self-supervised. But since the presented in-domain specialist approaches performed the best, we concentrated on these.
>
> To 3: Note that in our experience, this is often not the case: fine-tuning ImageNet on another upstream task/dataset does not always outperform using ImageNet alone. The demonstrated performance gains (of up to +11% for tasks with low number of samples) are of importance and are significant as is clearly shown in Table 3 and Figure 1. Could you please specify why you think these are not significant? These approaches improved state of the art as shown in Table 4, and the trends are consistent across wide ranges of training sample sizes as shown in Figure 2.
>
> To 4: We disagree with the statement that these are minor improvements and that the used hyperparameter sets limited the performance. We achieved state-of-the-art results in comparison to prior work when using the in-domain representations; and we improved upon transfer from ImageNet. Initially we experimented a lot with different settings for hyper parameters and observed that the presented set was sufficient for a broad set of problems. In this sense, one cannot speak here of poor training accuracy - we achieve state of the art results for these datasets. Potentially, the large scale of the BigEarthNet and So2Sat can be better utilized in the future, but it is not guaranteed and is subject to more research (note that both these datasets can be categorized as weakly supervised due to the nature of label acquisition, contrary to the other datasets which were hand-curated by humans).
> As discussed above and in the paper, we used the full datasets for upstream training, and evaluated the quality of the learned representations on small training sizes downstream (eg. 1000 or 100 training samples only, which is the critical area in remote sensing).
>
> To 5: We agree that this is a debatable point and therefore it is discussed in the paper. This was a surprise to us that RESISC-45 was outperforming other datasets for pre-training. One of the main reasons we believe is due to unique multi-scale aspect of this dataset (and we are going to explore it more in future work, applying it to other datasets as well).
> We disagree on some parts of the comment: the label spaces are quite different, but RESISC-45 still performs very well even for BigEarthNet and So2Sat, with which it does not share many categories/labels.
> We will explore other non-classification tasks in the future, but this is unfortunately out of scope for this work.
>
> To 6: There seems to be a misconception of what we tried to achieve with this paper. With this work, we are addressing a challenging domain, providing datasets, defining common benchmarks and demonstrating state-of-the-art performance based on various approaches. To our knowledge, this is the first generic and systematic study on remote sensing representation learning based on different tasks and datasets. We addressed the question of what remote sensing dataset characteristics are important, but we do not claim and did not want to make in impression that we solved it.
> Transfer from ImageNet provides a great baseline, and with this work we look into what kind of specialized data can be used to improve it further for the remote sensing domain. As demonstrated, we were able to significantly outperform ImageNet pre-training especially in the regime of low number of training samples.
> We explicitly want to explore diverse datasets with different characteristics to train more generalizable representations.

---

> > ### Author Response · Authors · 2019-11-14
> > **Continuation of the Response to Reviewer #3**
> >
> > To minor comments:
> >
> > - There are different ways to compute precision and recall metrics for multi-label classification problems, and the exact definition was not specified in the referenced work. As well, it is mentioned in the reference that about 12% of samples were affected by cloud and snow coverage obfuscating the actual scene and labels. In the reference, these samples were removed significantly simplifying the classification problem. Therefore, first, we cannot reconstruct the actual problem definition (which is one of the reasons we publish these standardized datasets and report results on standard splits), and, second, we consider our results on the available published dataset as a lower bound for the achievable accuracy (since we did not filter out the snow and cloud covered samples).
> >
> > - We train the representations using the full dataset upstream. Only for the downstream evaluation of the trained representations we use smaller number of training samples (100, 1000, or even 25) during the downstream training. We added a paragraph to the Experimental Results section to make this more clear.
> > Note that the accuracy change of BigEarthNet with finer training size steps (between 25 and 2500) is shown in Figure 2a), which as well clearly demonstrates the quite monotone/predictable trend and we expect this trend to continually flatten out and converge at some larger training sizes as is shown for the other smaller datasets.
> >
> >
> > Thank you for the suggestions.
> > Please note that we did not intend this paper to be a comprehensive survey, and we believe the presented results already constitute a useful contribution and discussion, and are of importance to the community.

---

### Official Review · AnonReviewer2 · 2019-11-04
**Official Blind Review #2**

**Rating:** 1

**Review:**

Summary:
The authors present a study on 5 remote sensing datasets where in-domain representation learning is studied and various methods to train and transfer knowledge between them are assessed. In addition, the authors aim to release these 5 datasets with appropriate training and test conditions to the public.
An interesting point of the paper is that ImageNet representations are used to either pretrain, finetune or train networks on each dataset and to transfer that performance to each other dataset. The performance of ImageNet is highly competitive int his context, demonstrating the quality of its features.
The paper is not proposing technical novelties of any kind throughout.

Comments:
-pixel real estate and resolution makes it unclear which dataset is bigger, though the authors attempt to discuss this. The 'winning' dataset (in terms of its usefulness for transfer) seems to have the highest resolution and image size, which may easily be the reason that models can better be trained here.
- Some of the tasks the authors attempt might best be attempted with different models. For example, not every remote sensing task is about classification of a patch. In some cases, segmentation into types of areas would be more appropriate. As such, the basic premise of the paper that the 5 tasks are similar and can be treated as one benchmarkable object seems quite brittle. A better approach seems to surely be to treat each problem with a more tailored approach?
Adding the right inductive biases into each problem may also help mitigate the lack of specific domain data when aiming for high performance in each domain.
- The authors miss a chance to consider this an exercise to perform a version of multitask learning for many of their experiments of similar structure, i.e. aerial images. In many cases satellite images will require similar RGB features. Why should we not use knowledge from all  datasets to get the best features? I was looking forward to seeing a discussion on this topic but unfortunately did not find it.

Decision:
Overall, this paper handles an interesting collection of data, but does not add much interesting novelty to the field of representation learning and fails to leverage some opportunities in those datasets to do sth. interesting (i.e. better transfer and/or better models).
I also find the approach to tackle all 5 datasets and all problems in each dataset as a similar problem too simplistic. Remote sensing data, as many real-world datasets, is a wonderful playground for rich modeling ideas and inductive biases to make things work with less data or particularly structured data.
It is commendable to prepare the datasets for publication and the basic results the authors show make for good first baselines here, but ultimately do not meet the bar for novelty or scientific insights.
The paper might be a stronger fit for a targeted workshop on its topic.

**Experience Assessment:**

I have published in this field for several years.

**Review Assessment: Checking Correctness Of Derivations And Theory:**

I carefully checked the derivations and theory.

**Review Assessment: Checking Correctness Of Experiments:**

I carefully checked the experiments.

**Review Assessment: Thoroughness In Paper Reading:**

I read the paper thoroughly.

---

> ### Author Response · Authors · 2019-11-14
> **Thank you for the review and insightful comments.**
>
> Thank you for the review and insightful comments.
>
> Concerning the novelty: we explore how much prior knowledge from in-domain datasets can help in general for other remote sensing problems, which has not been done before in this domain to our knowledge. We demonstrate that this single approach allows us to achieve state of the art on five different datasets without any dataset specific tuning.
>
> We agree that ImageNet provides very strong features, and the limited amount of improvement that we can achieve over it when using the full training data is a result in itself. Note that especially in the low training samples regime (which is of most importance to many remote sensing applications) our approach is a significant improvement over just using ImageNet.
>
> The image sizes, resolutions and dataset sizes are presented in Table 1. While this is an important point that the number of training samples is not a sufficient number to quantify the dataset size, please note that we used the same preprocessing for all datasets resulting in same input image sizes. The resolution is important in relation to the scale of the downstream tasks. Having only high-resolution upstream data for low-resolution tasks might be non ideal and vice versa. One of the insights of this paper was that multi-resolution helps in training better and wider applicable representations for remote sensing.
>
> We agree that different models could and should be used for many other downstream tasks and applications. However, in this paper we explicitly want to explore a standardized framework to evaluate representation learning on a simple problem (chip classification in this case). The 5 tasks are actually diverse in terms of label spaces, focus areas, and input image types. We specifically selected not to try to focus on a single or few tasks but to develop a general approach that we can evaluate on multiple tasks.
>
> One goal of this paper is to develop general representations that can be easily used by other remote sensing practitioners, and therefore these representations should be good for a wide range of downstream tasks and applications, avoiding the tailoring to a specific problem. For this goal we also want to open-source and publish the best trained representations in order for others to quickly try out on various tasks.
>
> Please note that while we did not account for inductive biases for individual datasets, we discuss and account for the general biases of remote sensing datasets, such us rotation variance or the importance of multi-scale.
>
> Thank you for the point about multi-task learning. This is something we want to explore, but it is out of scope for this paper.

---

### Decision · Program_Chairs · 2019-12-19

**Decision:**

Reject

**Comment:**

As the reviewers point out, this paper requires a major revision and fleshing out of the claimed contribution before it is suitable for conference presentation.